# Drop of Butyrylcholinesterase Activity after Cyclophosphamide Conditioning as a Predictive Marker of Liver Transplant-Related Complications and Its Correlation with Transplant-Related Mortality in Pediatric Hematopoietic Stem Cell Recipients

**DOI:** 10.3390/jcm8060825

**Published:** 2019-06-10

**Authors:** Natalia Maximova, Giulia Caddeo, Davide Zanon, Alessandra Maestro, Roberto Simeone

**Affiliations:** 1Bone Marrow Transplant Unit, Institute for Maternal and Child Health-IRCCS Burlo Garofolo, via dell’Istria 65/1, 34137 Trieste, Italy; 2Department of Medicine, Surgery and Health Sciences, University of Trieste, Piazzale Europa 1, 34127 Trieste, Italy; g.caddeo88@gmail.com; 3Pharmacy and Clinical Pharmacology Unit, Institute for Maternal and Child Health-IRCC Burlo Garofolo, via dell’Istria 65/1, 34137 Trieste, Italy; davide.zanon@burlo.trieste.it (D.Z.); alessandra.maestro@burlo.trieste.it (A.M.); 4Department of Transfusion Medicine, ASUITS, Piazza dell’Ospitale 1, 34129 Trieste, Italy; roberto.simeone@asuits.sanita.fvg.it

**Keywords:** serum butyrylcholinesterase activity, cyclophosphamide, pediatric transplant recipients, liver transplant-related complications, defibrotide

## Abstract

Transplant-related liver complications are a potentially fatal condition of hematopoietic stem cell transplantation (HSCT) in pediatric patients, actually representing one of the main factors involved in transplant-related mortality (TRM). The search for a specific marker capable of predicting the development of this condition is a relevant clinical issue. We have observed a variable reduction in serum butyrylcholinesterase (BChE) activity after a cyclophosphamide-containing conditioning regimen. This study aims to determine the cutoff of BChE activity reduction that might be a specific prognostic marker for liver complications after HSCT. Our results show that the reduction of BChE values below 2000 U/L the day before the transplantation is an indicator strongly associated with the transplant-related liver complications (*p* < 0.0001). The incidence of overall survival at 1 year was significantly higher in the BChE > 2000 U/L group compared to the BChE < 2000 U/L group (84.7% versus 58.5%, *p* < 0.001), while the TRM rate was significantly lower (8.1% versus 23.1%, *p* < 0.05). None of the patients undergoing prophylaxis with defibrotide developed severe liver complications. Starting defibrotide treatment at the first signs of hepatic dysfunction in patients with particularly low BChE activity levels reduces severe liver transplant-related complications.

## 1. Introduction

Hematopoietic stem cell transplantation (HSCT) has become a well-established treatment for various malignant and non-malignant disorders originating from the hematopoietic system [1].

Improvements in transplantation techniques over the recent years have led to a significant reduction in treatment-related complications.

Different areas of improvement have included changes in conditioning regimens, larger donor availability, better Human Leucocyte Antigens (HLA)—matching techniques and graft source manipulation, advanced techniques to detect minimal residual disease, posttransplant immunotherapy, progress in supportive care, graft-versus-host disease (GVHD) prophylaxis and therapy, detection and treatment of subsequent infections [2].

Despite the reduction in overall transplant-related mortality (TRM) over time, it currently exceeds 10% in allogeneic settings [3]. Indeed, it depends above all on the relative stability in the incidence of liver-related complications, with approximately 70% of pediatric patients displaying abnormal levels of either aminotransferases or bilirubin [4]. Previous studies have shown how pretransplant liver dysfunction is related to higher risk of overall TRM. Factors involved in this dysfunction include chronic viral hepatitis, pretransplant chemotherapy-induced liver toxicity, and severe iron overload from multiple transfusions [5,6,7,8]. Clinically significant liver disease presenting early after HSCT, as in the case of sinusoidal obstruction syndrome (SOS) or acute GVHD, plays an important role in determining adverse outcomes, being usually associated with poor survival and often considered to be the primary cause of death [9].

SOS, also known as veno-occlusive disease (VOD), is a potentially fatal complication of HSCT and a relevant challenge in the management of children undergoing a transplantation.

Its incidence is quite variable depending on the composition and intensity of the conditioning regimen. In the pediatric setting, most of the conditioning regimes are myeloablative and includes the use of cyclophosphamide and total body irradiation (TBI).

Reported rates of SOS in patients receiving this conditioning regime often exceed 50% [10]. While mild or moderate SOS usually resolves within weeks in most patients, the severe form, which is estimated to involve 30–60% of affected children, is associated with multiorgan dysfunction and a mortality rate higher than 80% [11,12].

The transplant community aims to achieve a further reduction in TRM after allogeneic HSCT, bringing it down to 5%. Reducing TRM due to liver dysfunction is a way to achieve this goal.

Butyrylcholinesterase (BChE), also known as pseudocholinesterase, is a major detoxification enzyme capable of hydrolyzing many different choline-based esters. BChE is synthesized in the liver and, for this reason, it is commonly used in the clinical routine as a quick and cheap biomarker for liver function, reflecting hepatic synthetic capacity. Lower BChE activity levels correlate with a more extensive liver injury [13] and are associated with disease severity and mortality in critically ill patients [14,15,16], especially with respect to systemic inflammation and sepsis [17,18].

In our 15 years of experience with butyrylcholinesterase (BChE) activity monitoring, we have observed a reduction in the serum BChE activity levels in all patients given a pretransplant conditioning regime containing high-dose of cyclophosphamide. The decline in serum BChE activity values was variable in each patient, from a minimum of 20% to a maximum of more than 80%.

The primary objective of this study is to determine the association between the degree of BChE activity reduction after two days of cyclophosphamide treatment and the development of transplant-related liver complications after HSCT. The secondary aim is to define a cutoff of BChE activity decrease which allows identification of those patients who are at high risk for liver complications and to eventually tailor the therapeutic approach.

## 2. Methods

A retrospective single-center study was conducted at the pediatric transplant center of the Institute for Maternal and Child Health “IRCCS Burlo Garofolo” in Trieste, Italy. The study protocol was approved by the institutional review board of the institute (reference no. 10\2018).

All patients have given written consent for collection and use of personal data.

### 2.1. Patients

Medical records of all patients who underwent allogeneic or autologous HSCT at our center between January 2003 and January 2018 were investigated.

Inclusion criteria in the study were: Age of recipient under 18 years at the time of transplantation, first transplant attempt, use of a myeloablative conditioning regime including high-dose cyclophosphamide, documented serum BChE activity levels during conditioning. All patients aged over 18 years at the time of transplantation were excluded. Disease risk was established considering the type of diagnosis and the disease stage [19]. Patients with genetic diseases, aplastic anemia, refractory cytopenia, and leukemia in first or second remission were included in the low disease risk group, while patients with solid tumors, relapsed leukemia (or leukemia in more than second remission), and refractory anemia with excess blasts were considered at high risk.

### 2.2. HSCT Procedure

All patients were treated according to standard myeloablative protocols. In patients over 2 years of age with acute lymphoblastic leukemia (LLA), the myeloablative conditioning regime preceding allogeneic HSCT was based on total-body irradiation (TBI), while in the remaining cases, a busulfan-based conditioning regime was used. In both cases, conditioning also included high-dose cyclophosphamide (1800 mg/m^2^ for two consecutive days). In the case of matched unrelated donors, haploidentical or sibling donors and patients with hemoglobinopathy, rabbit anti-thymocyte globulin (ATG) was used. GVHD prophylaxis was performed with calcineurin-inhibitor alone or associated with mycophenolate mofetil and prednisone, as previously described [20].

All patients undergoing an autologous HSCT received busulfan orally (360 mg/m^2^) and cyclophosphamide (1800 mg/m^2^ for two consecutive days). In case of abnormal bilirubin values, during both autologous and allogeneic HSCT, the second dose of cyclophosphamide was slightly reduced.

### 2.3. Liver Disease at Transplant Assessment

Before 2012, the level of liver siderosis was estimated by calculating the number of blood units transfused and determining the serum ferritin concentration [21]. Liver iron overload assessment by liver biopsy was performed in only 14% of the patients. Beginning in 2012, the liver iron concentration has been evaluated using magnetic resonance (MR) in all patients for whom HSCT is anticipated. The algorithm for quantification of iron concentration in the liver by MRI was previously described [22]. The diagnosis of all other pretransplant liver dysfunctions was histological.

### 2.4. Prophylaxis of Transplant-Related Liver Complications

Between January 2010 and June 2014, all patients receiving an allogeneic transplantation underwent SOS prophylaxis with defibrotide, while beginning in July 2014, prophylaxis has been limited only to high-risk patients. Defibrotide was started on the first day of conditioning and lasted until 28 days after HSCT, at a dose of 25 mg/kg divided into four administrations per day. Moreover, all recipients, in both autologous and allogeneic transplants, received prophylaxis with ursodeoxycholic acid (UDCA), at a dose of 30 mg/kg/day for a minimum of two months after HSCT, regardless of their liver function. In addition, the patients with abnormal liver functional tests (LFTs) received N-acetylcysteine at a dose of 150 mg/kg in single daily administration.

### 2.5. Criteria for the Diagnosis of Transplant-Related Liver Complications

The diagnosis of SOS was established according to the modified Seattle diagnostic criteria [23]. Posttransplant liver dysfunction was defined as SOS based on clinical symptoms including weight gain, hepatomegaly, ascites, and elevated bilirubin. All cases of mild, moderate, or late-onset SOS were histologically proven. Transplant-related liver damage was considered to have occurred only when histologically proven or in the presence of increased bilirubin and transaminases values with symptoms not fulfilling SOS diagnostic criteria. Liver damage was defined as mild in the case of bilirubin values between 2 and 3 mg/dL and transaminases levels not exceeding twice the normal range. Moderate damage was defined by the presence of bilirubin values from 3 to 5 mg/dL and transaminases values from 2 to 5 times higher than normal. Severe liver damage was defined by bilirubin values higher than 5 mg/dL, and transaminases values 5 times higher than normal. Clinical diagnosis and histological features of hepatic GVHD were based on the National Institute of Health (NIH) criteria [9,24]. In particular, histological features of acute liver GVHD were defined as the presence of dysmorphic or destroyed small bile ducts with or without cholestasis and lobular and/or portal inflammation.

### 2.6. Statistical Analysis

Collected data were analyzed using descriptive statistics to determine the distribution and frequency of the variables. Continuous variables were expressed as median and confidence interval (CI) between second and third quartiles (percentile 25 and percentile 75), while categorical variables were expressed as the frequency, and absolute or a percentage value. Box and whisker plots were generated for displaying the distribution of the numeric variable. The Mann–Whitney test was used to compare the different groups of patients as appropriate. The two-tailed Fisher exact test was performed to assess the association between categorical variables. We assessed the validity of biochemical parameters in predicting the transplant-related liver complications by evaluating the respective area under curve (AUC) and receiver operating characteristic (ROC) curves. The Youden index was used to establish the best cutoff for sensitivity and specificity of each variable. Kaplan–Meier plots were generated for a graphical explanation of clinical outcomes. *p*-values < 0.05 were considered as statistically significant. The Cox proportional-hazards regression model was used to investigate the association between the survival time of patients and a series of possible predictive variables. Statistical analyses were performed using WinStat (v.2012.1; In der Breite 30, 79189 Bad Krozingen, Germany) and MedCalc (Statistical Software version 18.9.1, Ostend, Belgium; http://www.medcalc.org; 2018).

### 2.7. Analysis of Biochemical Parameters

Biochemical parameters such as alanine aminotransferase (ALT), aspartate aminotransferase (AST), gamma-glutamyltransferase (GGT), BuChE, total bilirubin, and C reactive protein (CRP) concentrations were measured by AU Beckman Coulter reagents and autoanalyzer (Indianapolis, IN, USA).

## 3. Results

### 3.1. Demographic Features

During the study period (from January 2003 to January 2018) 225 HSCTs in 214 patients were performed in our transplant center. The eligibility criteria was met by 176 pediatric patients, 110 boys and 66 girls, for whom this was their first HSCT. At the time of analysis, 75% of the patients were alive, while 25% had died. The minimum follow-up for survivors was 12 months. Median age at the time of transplantation was 9 years (range 0–17).

Table 1 summarizes the baseline patient demographics. As shown, acute lymphoblastic leukemia was the most common underlying disease treated, followed by non-malignant disorders, acute myeloid leukemia, myelodysplastic syndromes, and malignant solid tumors. Non-malignant disorders consisted of hemoglobinopathy, primary immunodeficiencies, inherited metabolic diseases, and osteopetrosis. Solid tumor diagnoses included neuroblastomas, Ewing sarcomas, primitive neuroectodermal tumors, and rhabdomyosarcomas.

The majority of patients had a low disease risk at transplant (62% versus 38% in the high disease risk group). All 176 HSCTs, 92% allogeneic and 8% autologous, were preceded by a myeloablative conditioning regime. Chemotherapy-based conditioning was more frequently used compared to TBI-based conditioning (64% versus 36%). The mean total dose of cyclophosphamide received during conditioning was only slightly lower compared to the standard dose (3500 mg/m^2^ versus 3600 mg/m^2^).

One hundred fifty-three patients (87%) had liver disorders at transplant with a clear predominance of liver siderosis (72%).

### 3.2. Analysis of the Suitability of the Prognostic Markers Chosen for the Study

First of all, we evaluated a possible relationship between baseline serum BChE activity levels and subsequent severe transplant-related liver disorders (SOS and third-grade liver disease). As shown in the corresponding ROC curve (Figure 1A), baseline BChE activity values were very poor in predicting the onset of transplant-related liver damage (AUC = 0.507; 95% confidence interval (CI) = 0.49–0.64; *p* = 0.932). The maximum Youden index for absolute BChE activity values was 6820 U/L with a sensitivity of 55.2% and a specificity of 63.3%. In an analogous way, we evaluated the predictive performance of the absolute BChE activity values on day −1 (Figure 1B). We observed that a serum BChE activity value ≤ 1799 U/L performed best in predicting a severe posttransplant liver damage with AUC = 0.801 and 95% CI = 0.73–0.86 (*p* < 0.001), sensitivity 72.4% and specificity 80.8%.

Considering only the patients who underwent allogenic HSCT for hematologic malignancies, BChE activity value ≤ 1799 U/L maintained a good predictive performance for severe liver damage with AUC = 0.761 and 95% CI = 0.673–0.835 (*p* < 0.001), sensitivity 63.2% and specificity 84.7%.

In addition, we evaluated the performance of some other commonly used markers such as C-reactive protein (CRP), total bilirubin, alanine aminotransferase (ALT), aspartate aminotransferase (AST), gamma-glutamyltransferase (GGT) on day −1, obtaining the respective ROC curves. None of these variables showed acceptable predictive performance (Table 2).

Preliminary analysis was performed in order to assess if the correlation between BChE activity and each of the other markers listed in Table 2 could improve the predictive performance but the test did not show significative results.

### 3.3. Analysis of the Relationship between Patients’ Variables and the Reduction in the Serum BChE Activity Level

In all 176 patients, we detected a variable reduction in serum BChE activity values ranging from 13.9% to 92.1%. The most significant BChE activity drop was documented on day −1, after cessation of cyclophosphamide (Figure 2). We considered both the absolute and percentage value of BChE activity reduction on day −1 and we divided the study cohort into two groups, respectively. Percentage BChE activity loss was calculated by comparing the baseline values with those documented on day −1. The results are listed in Table 3 and Table 4, respectively.

The cutoff point was arbitrarily set at 2000 U/L in the case of absolute reduction, and at 70% in the case of percentage reduction. Cutoff of 2000 U/L, chosen due to sample size, is very close to maximum Youden index of 1799 U/L. The BChE activity basal values were significantly higher in the BChE > 2000 U/L group, compared with the BChE < 2000 U/l group (*p* < 0.05), and in the BChE drop > 70% group compare with the BChE drop < 70% group (*p* < 0.001).

Statistical analysis showed that the degree in BChE activity level reduction was not associated with patient demographic features such as, gender, age at transplant, ferritin baseline level, presence of liver disease at transplant, etc. It was also independent of any liver toxicity prophylaxis.

Considering the primary disease, only an association between acute lymphoblastic leukemia and BChE drop > 70% (*p* < 0.001) was detected. The high disease risk had a close correlation with BChE activity loss (*p* < 0.0001) and this association was seen also for those patients treated with TBI conditioning (*p* < 0.05 Table 2; *p* < 0.001 Table 3).

In addition, severe iron overload was significantly related to BChE loss (*p* < 0.05), and abnormal baseline ferritin levels showed high predictive performance for the onset of transplant-related liver damage (AUC = 0.691; 95% CI = 0.62−0.76; *p* < 0.001). The maximum Youden index was 1028 μg/L for ferritin baseline level, with a corresponding sensitivity of 86.2% and specificity of 53.8%.

### 3.4. Analysis of the Relationship between the Reduction in Serum BChE Activity Level and the Transplant Outcomes

We looked at the association between the decrease in the absolute serum BChE activity values and the onset of transplant-related liver complications (Table 5). The statistical analysis showed that BChE fall < 2000 U/L was associated with the onset of posttransplant liver dysfunction and SOS (*p* < 0.05). No statistically significant differences in the cumulative incidence of any grade of GVHD (in both the BChE > 2000 U/L and BChE < 2000 U/L groups) were observed. In a follow-up after 12 months, the incidence of overall survival (OS) was significantly higher in the BChE > 2000 U/L group compared with the BChE < 2000 U/L group (84.7% versus 58.5%, *p* < 0.001). In total, 44 patients (25%) died, 24 (13.4%) for transplant-related complications and 20 (11.4%) for the recurrence of the underlying disease. Transplant-related deaths occurred at a median of 91.7 ± 75.6 days, while disease recurrence deaths at a median of 211.3 ± 78.0 days. The cumulative mortality rate was significantly higher in the BChE < 2000 U/L group compared with the BChE > 2000 U/L group (41.5% versus 15.3%, respectively, *p* < 0.001), with the same trend observed both for TRM (23.1% versus 8.1%, *p* < 0.05) and disease recurrence mortality (18.5% versus 7.2%, *p* < 0.05). BChE activity drop < 2000 U/L was significantly associated with a higher probability of transplant-related death (Odds Ratio (OR) = 3.4; 95% CI = 1.39−8.3; *p* < 0.005). Two patients died due to VOD in the BChE > 2000 U/L group versus nine patients in the BChE < 2000 U/L; five and four patients, respectively, died due to infections; one and two patients, respectively, died due to acute GVHD; and one patient in the BChE > 2000 U/L group died due to diffuse alveolar hemorrhage.

Analyzing the association between the percentage reduction in BChE activity and the transplant outcomes (Table 6) we found that overall survival (OS), TRM, and disease recurrence related death were not associated with BChE percentage decrease >70%. Considering all liver dysfunctions, only SOS had a strong association with BChE percentage drop (*p* < 0.005).

We evaluated the predictive performance of both absolute and percentage BChE values on day −1 in the diagnosis of SOS, obtaining the respective ROC curves (Figure 3A,B), which showed how the absolute and percentage BChE values performed at similarly high performances (AUC = 0.76; 95% CI = 0.69–0.82; *p* < 0.001). In trying to identify a cutoff of absolute BChE value on day −1, we found the maximum Youden index was 1571 U/L with specificity of 82.9% and sensitivity of 62.5% for the diagnosis of SOS. Regarding the percentage value, a cutoff of 71.6% correlated to a specificity of 62.9%, although with a sensitivity of 87.5%.

We looked at the association between the absolute BChE values on day −1 and 1 year posttransplant follow-up outcomes such as, overall survival (OS) and TRM rate. One year OS in the BChE > 2000 U/L group was 84.7% versus 58.5% in the BChE < 2000 U/L group (*p* < 0.001), while TRM was 8.1% versus 23.1% (*p* < 0.05), respectively.

Kaplan–Meier curve analysis confirmed the statistically higher survival probability (*p* < 0.0001) (Figure 4A) and lower probability of TRM (*p* < 0.0001) in the BChE > 2000 U/L group (Figure 4B).

The apparent increase of mortality between days 0–70 and days 150–250 reflects the common trend of early and late TRM. Early forms are mainly caused by acute GVHD, infections, SOS, and other angiopathic diseases, while late forms by chronic GVHD, pulmonary fibrosis, secondary tumors, and hepatic or kidney insufficiency.

An analogue Kaplan–Meier analysis was performed to assess TRM considering only patients who underwent an allogenic HSCT for hematologic malignancies. The analysis showed a statistically lower TRM probability (*p* < 0.05) in the BChE > 2000 U/L group.

Finally, using a Cox proportional-hazards regression model, we investigated the association between TRM and some other variables including type of diagnosis, disease risk, source of stem cells, year of transplantation, type of conditioning, preexisting liver siderosis, or other pretransplant hepatopathy. Both in the BChE > 2000 U/L and in the BChE < 2000 U/L groups, the only variable that turn out to be associated with TRM was disease risk (*p* < 0.0001).

### 3.5. Analysis of Patient and Disease Variables Associated with the Risk of Severe Liver Damage Development

We studied the possible association between some patient and disease variables and the transplant-related severe liver damage development (Table 7). Of the 176 pediatric patients analyzed, 44 (25%) developed severe liver damage, while 132 (75%) developed no impairment or only a mild grade damage. The patients’ underlying disease and conditioning regimen were not correlated with the risk of liver damage.

However, we found that patients with a low risk disease had a lower probability of developing severe liver damage during the follow-up period (27.3% versus 72.7%, *p* < 0.0001). Severe iron overload and higher serum ferritin levels proved to be related in a statistically significant way to the probability of developing severe liver damage (*p* < 0.0001 and <0.05, respectively). Among all patients who underwent SOS prophylaxis with defibrotide, we did not identify any case of severe liver damage (*p* < 0.0001). As a result, overall survival was significantly higher for patients in which liver damage was absent or mild compared with those who had severe liver damage (85.6% versus 43.2%, *p* < 0.0001).

## 4. Discussion

Cyclophosphamide is a chemotherapeutic agent widely used in various combinations with other drugs for the treatment of a large number of different malignancies. It is also included in the vast majority of conditioning regimes before HSCT [25]. Hepatic microsomal enzymes convert cyclophosphamide into active cytotoxic metabolites. In particular, the mixed function oxidase system (cytochrome P450) converts cyclophosphamide in 4-hydroxy-cyclophosphamide and aldophosphamide [26]. These metabolites are then transported to the healthy and neoplastic tissues where aldophosphamide is converted to the alkylating phosphoramide mustard and acrolein, which produces reactive oxygen species (ROS) with subsequent oxidative tissue damage. Acrolein also induces cellular damage mediated by lipid peroxidation [27]. Oxidative stress created by cyclophosphamide metabolites and the direct toxicity of aldehydes are responsible for acute tissue toxicity, and in particular of hepatic toxicity [28]. The association between the administration of high-dose cyclophosphamide and a significative reduction in BChE levels has been previously described [29,30,31,32]. The underlying mechanism is not clear, but the rapidity of enzyme inhibition in vitro suggests a direct effect rather than an interference with BChE synthesis [33]. At the same cyclophosphamide dose, the suppression in BChE activity varies widely from patient to patient, and in some cases, can last for several days. Despite these findings, the possible association between the basal value of BChE activity or its reduction with therapy and the incidence of short-term transplant-related liver complications has been never investigated.

In our study, we analyzed the performance of various commonly used markers such as CRP, total bilirubin, ALT, AST, GGT, in predicting the probability of hepatic transplant-related complication development. Considering their baseline values obtained before starting the conditioning regime, none of these variables showed acceptable predictive performance. Regarding BChE activity level, we found that the baseline values had little predictive significance, while the same measurement obtained on day −1 had high sensitivity and high specificity in predicting hepatic transplant-related complications.

Our analysis shows how the most relevant factors predisposing significant BChE activity reduction are a high disease risk, the use of TBI-based myeloablative conditioning and a severe preexistent iron liver overload. Some of these findings were predictable. Patients with an underlying high-risk disease correspond to those with an advanced stage of pathology which usually had already been subjected to different courses of therapy. In this group of patients, the complexity of the therapies leads to an expected increased drug-induced liver toxicity, as witnessed by the lower basal BChE activity values recorded. In an analogous way, TBI-based conditioning and preexisting iron overload are a well-known risk factor for SOS [34,35,36,37,38], so it is not surprising that BChE reduction is more marked in these groups. Looking at the percentage reduction in BChE activity values, we found that a more significant decrease was associated with the diagnosis of LLA. This data is predictable since TBI is the conditioning choice in this type of patient.

In terms of predictive performance, our data show the superiority of the absolute BChE activity values compared to the percentage ones. This is mainly due to the significantly higher basal BChE activity level of patients in whom a reduction >70% was detected compared with those included in the group who reached a value <2000 UI/L. In fact, in the first case, a substantial percentage reduction >70% led, in some patients, to absolute values within the limits of the standard range while, in the second, a lower percentage reduction led to very much lower BChE activity reflecting impaired hepatic synthesis.

Another relevant finding that emerged in our study is the association between BChE activity reduction and the development of hepatic non-immunologic complications due to direct drug-induced or radiotherapy-induced hepatic damage. In contrast, we have not found any association between BChE activity and the incidence or severity of hepatic GVHD.

It is well known that disease risk, iron overload, ferritin levels, and the presence of preexisting liver disease are all factors correlated with TRM [35,39,40]. In our analysis, the pretransplant liver disease did not prove to be associated with hepatic complications, but this finding is probably due to the low sample size.

Our data confirmed the efficacy of defibrotide administration in the treatment of SOS, with a significant impact in reducing TRM and improving OS. Defibrotide is a complex mixture of polydeoxyribonucleotides with strong protective and anti-inflammatory activities on the vessel endothelium [41,42]. We used it in prevention for a few years in our transplant unit, zeroing SOS incidence in that period. This result is very encouraging considering that the incidence rate of SOS reported in the literature is around 10–15% in allogeneic HSCT recipients after a myeloablative conditioning regime, with the mortality rate in patients with severe SOS still exceeding 80% and 20% in patients with non-severe SOS [43,44]. Unexpectedly, our data showed the increased incidence of relapse in the group of patients treated with defibrotide. This phenomenon is probably due to a significant increase in OS in the defibrotide-treated group.

So far, SOS represents the main factor involved in TRM. The diagnosis is not always easy since signs and symptoms of SOS may be very insidious, particularly in pediatric patients, or may be confused with other conditions [11]. Early administration of defibrotide is the only effective treatment in the management of SOS, but currently, its use is formally indicated only in cases of severe SOS [45,46]. The prophylactic use of defibrotide is presently off-label. Moreover, its high cost represents a further limitation to its prescription in many centers.

Identification of markers with high sensitivity and specificity in predicting the risk of SOS is a relevant clinical issue. These selective markers would allow identification of those patients who can benefit from early treatment with defibrotide and to further improve their prognosis. In the past decades, numerous serum and plasma proteins have been investigated [47]. Plasminogen activator inhibitor-1 (PAI-1) is very sensitive, with a negative predictive value of around 100%, but nonspecific, since it does not discriminate between different causes of liver dysfunction [48]. Moreover, the measuring of PAI-1 is not feasible everywhere nor in a real time, and this strongly limits its use in clinical practice. Ultrasonography, computerized tomography, and magnetic resonance imaging may yield abnormal results in patients with SOS [49], but they are unlikely to be used as diagnostics in the early stage of the disease. Liver stiffness measurements with transient elastography have been recently considered as a promising technique in the early diagnosis of SOS [50]. However, elastography has some limitations, such as the high cost of the device and operator-related reliability.

BChE activity level is an inexpensive and quick marker feasible in any laboratory. Its degree of reduction during myeloablative conditioning including cyclophosphamide, correlates with the risk of subsequent SOS development and can be used to identify the high-risk patients.

In conclusion, we suggest that it could be reasonable to start a prophylactic therapy with defibrotide at the first signs of hepatic dysfunction, even before official diagnostic criteria for SOS are met, in patients at high risk for liver complications in whom a drop in BChE activity values <1800 UI/L is detected.

To our knowledge, this is the first study addressing the association between BChE reduction during conditioning regimen and transplant outcomes with particular attention to liver damage.

However, this study has some limitations that should be considered when interpreting the current findings: First of all, the sample size was relatively small and the follow-up was short term; second, the study was retrospective and monocentric in its nature.

Further research with a prospective study should be pursued to determine the role of BChE with particular attention to hepatic damage in patients undergoing a conditioning regimen with cyclophosphamide and its association with transplant outcomes.

## Figures and Tables

**Figure 1 jcm-08-00825-f001:**
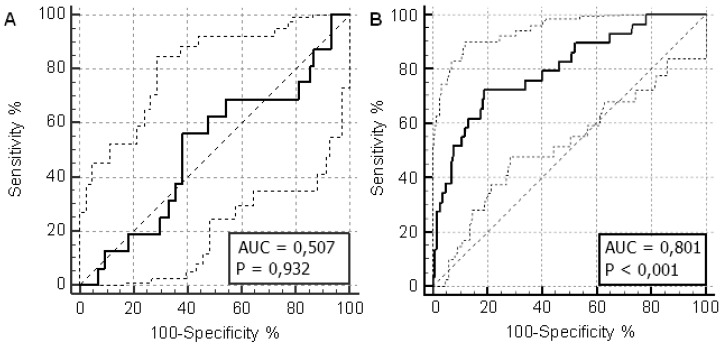
Diagnostic performance of butyrylcholinesterase (BuChE) activity in predicting the onset of transplant-related liver damage. Receiver operating characteristic (ROC) curves for absolute baseline BuChE activity values (**A**) and absolute BChE activity values on day −1 (**B**).

**Figure 2 jcm-08-00825-f002:**
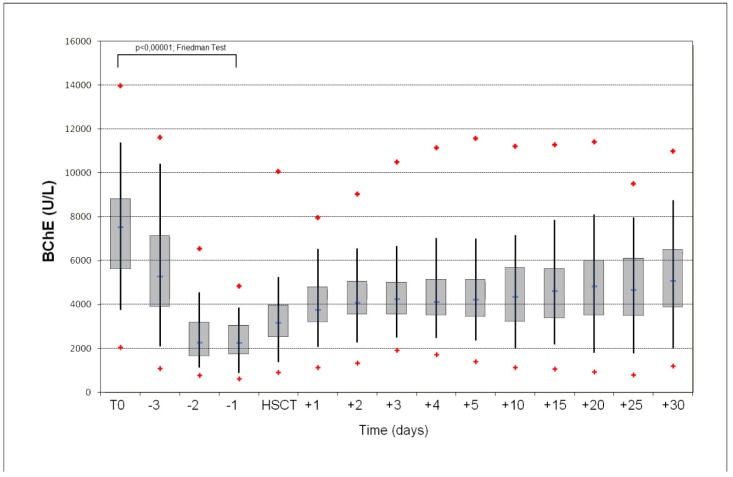
Box-and-whisker plots show a trend of butyrylcholinesterase (BuChE) activity during and after cyclophosphamide treatment.

**Figure 3 jcm-08-00825-f003:**
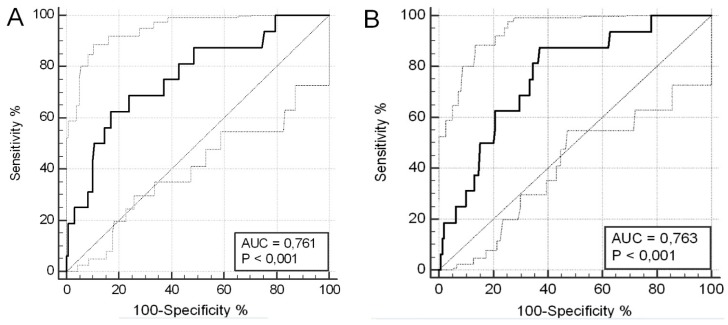
Predictive performance of butyrylcholinesterase (BuChE) activity on day −1 in the diagnosis of sinusoidal obstruction syndrome (SOS). Receiver operating characteristic (ROC) curves for absolute BuChE activity values (**A**) and percentage reduction in BChE activity (**B**).

**Figure 4 jcm-08-00825-f004:**
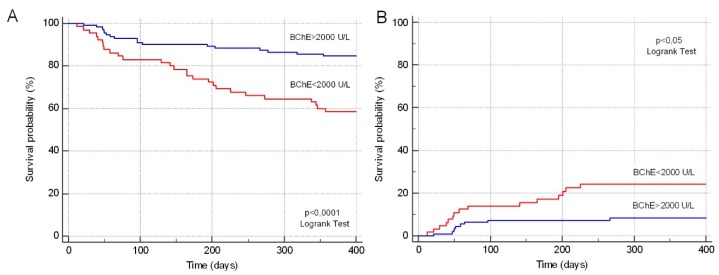
Kaplan–Meier curve analysis of the survival probability (**A**) and transplant-related mortality (**B**).

**Table 1 jcm-08-00825-t001:** Patient demographics.

Pre-Transplant Baseline Characteristics	Whole Cohort
**Number of patients (%)**	176 (100)
**Sex:**	
Male (%)	110 (62.5)
Female (%)	66 (37.5)
**Age at transplant, years (mean [± SD])**	9.2 (5.1)
**Underlying disease, number (%):**	
Acute lymphoblastic leukemia	65 (36.9)
Acute myeloid leukemia	33 (18.8)
Myelodysplastic syndrome	22 (11.9)
Solid tumour	22 (11.9)
Nonmalignant disorders	34 (19.3)
**Disease risk, number (%): ***	
Low	109 (61.9)
High	67 (38.1)
**Myeloablative conditioning, number (%):**	
MCHT-based	113 (64.2)
TBI-based	63 (35.8)
**Cyclophosphamide total dose, mg/m^2^, (mean [± SD])**	3499.4 (217.7)
**Type of transplantation, number (%):**	
Allogeneic	162 (92.0)
Autologous	14 (8.0)
**Graft source, number (%):**	
Bone marrow	127 (72.2)
Peripheral blood stem cells	39 (22.2)
Umbilical cord blood	10 (5.7)
**Allogeneic donor type, number (%):**	
Matched related donor	51 (31.5)
Matched unrelated donor	77 (47.5)
Haploidentical donor	34 (21.0)
**Liver disorders at transplant, number (%):**	
Siderosis	126 (71.6)
Viral hepatitis	11 (6.3)
Metabolic liver disease	11 (6.3)
Autoimmune hepatitis	5 (2.8)

MCHT, myeloablative chemotherapy; TBI, total body irradiation; SD, standard deviation. ***** Disease risk was defined according to previously published classification.

**Table 2 jcm-08-00825-t002:** Diagnostic performance of serum biochemical markers in predicting transplant-related liver dysfunction.

Biochemical Markers *	AUC—ROC (95% CI)	ROC—Significant Value (*p*)	Maximum Youden Index (Cut-Off)	Sensitivity (%)	Specificity (%)
**C-reactive protein (mg/dL)**	0.591 (0.514–0.664)	NS	0.5	86.2	34.7
**γGT (U/mL)**	0.607 (0.530–0.681)	NS	77	27.6	93.7
**ALT (U/mL)**	0.598 (0.521–0.672)	NS	28	55.2	65.5
**AST (U/mL)**	0.554 (0.477–0.630)	NS	20	62.1	57.2
**Total bilirubin (mg/dL)**	0.583 (0.506–0.657)	NS	0.29	100	22.1
**BChE (U/L)**	0.800 (0.733–0.857)	<0.0001	1799	72.4	80.8

* Biochemical markers evaluated the day before HSCT (day −1). γGT, γ-glutamyltranspeptidase; ALT, alanine aminotransferase; AST, aspartate aminotransferase; BChE, butyrylcholinesterase; AUC-ROC, area under the receiver operating characteristic curve; CI, confidence interval.

**Table 3 jcm-08-00825-t003:** Patient characteristics associated with drop of BChE absolute values on day −1.

Variables	BChE > 2000 U/L	BChE < 2000 U/L	*p*-Value *
**Number of patients (%)**	111 (63.1)	65 (36.9)	-
**Gender:**			
Male (%)	73 (65.8)	41 (63.1)	NS
Female (%)	38 (34.2)	24 (36.9)	-
**Age at transplant, (years, median [IQR])**	9 (4–13)	9 (5–14)	NS **
**Underlying disease, number (%):**			
Acute lymphoblastic leukemia	36 (32.4)	29 (44.6)	NS
Acute myeloid leukemia	25 (22.5)	8 (12.3)	NS
Myelodysplastic syndrome	16 (14.4)	6 (9.2)	NS
Solid tumor	11 (9.9)	11 (16.9)	NS
Nonmalignant disorders	23 (20.7)	11 (16.9)	NS
**Disease risk, number (%):**			
Low	86 (77.5)	24 (36.9)	<0.0001
High	25 (22.5)	41 (63.1)	-
**Myeloablative conditioning, number (%):**			
MCHT-based	78 (70.3)	35 (53.8)	<0.05
TBI-based	33 (29.7)	30 (46.2)	-
**Liver iron overload, number (%):**			
Absent	34 (30.6)	16 (24.6)	NS
Mild	22 (19.8)	9 (13.8)	NS
Moderate	22 (19.8)	8 (12.3)	NS
Severe	33 (29.7)	32 (49.2)	<0.05
**Baseline BChE value, (U/L, median [IQR])**	7789 (6028–9218)	6780 (4889–8031)	<0.05 **
**Serum ferritin, (μg/L, median [IQR])**	989 (343–2012)	1680 (402–2877)	NS **
**Liver disease, number (%)**	16 (14.4)	17 (26.1)	NS
**Defibrotide prophylaxis, number (%)**	41 (36.9)	27 (41.5)	NS
**UDCA prophylaxis, number (%)**	109 (98.2)	60 (92.3)	NS
**HDNAC prophylaxis, number (%)**	72 (64.8)	39 (60.0)	NS

Ferritin normal range 3.0–88.0 μg/L. IQR, interquartile range; UDCA, ursodeoxycholic acid; HDNAC, high-dose N-acetylcysteine. * Fisher’s test. ** *U*-Test (Mann–Whitney).

**Table 4 jcm-08-00825-t004:** Patient characteristics associated with percentage drop of BChE values on day −1.

Variables	BChE Drop < 70%	BChE Drop > 70%	*p*-Value *
**Number of patients (%)**	94 (53.4)	82 (46.6)	-
**Gender:**			
Male (%)	66 (70.2)	48 (58.5)	NS
Female (%)	28 (29.8)	34 (41.5)	-
**Age at transplant, (years, median [IQR])**	7 (4–13)	10 (6–13)	NS **
**Underlying disease, number (%):**			
Acute lymphoblastic leukemia	26 (27.7)	39 (47.6)	<0.001
Acute myeloid leukemia	20 (19.7)	13 (9.7)	NS
Myelodysplastic syndrome	16 (17.0)	6 (7.3)	NS
Solid tumor	12 (12.8)	10 (12.2)	NS
Nonmalignant disorders	20 (19.7)	14 (17.1)	NS
**Disease risk, number (%):**			
Low	67 (71.3)	34 (41.5)	<0.0001
High	27 (28.7)	48 (58.5)	-
**Myeloablative conditioning, number (%):**			
MCHT-based	71 (75.5)	42 (51.2)	<0.001
TBI-based	23 (24.4)	40 (48.8)	-
**Liver iron overload, number (%):**			
Absent	29 (30.9)	21 (25.6)	NS
Mild	19 (20.2)	12 (14.6)	NS
Moderate	20 (21.3)	10 (12.2)	NS
Severe	26 (27.7)	39 (47.6)	<0.05
**Baseline BChE value, (U/L, median [IQR])**	6815 (5115–8361)	7897 (6706–9408)	<0.001 **
**Serum ferritin, (μg/L, median [IQR])**	950.1 (322–2135)	1474.4 (406–2440)	NS **
**Liver disease, number (%):**	19 (20.2)	16 (19.5)	NS
**Defibrotide prophylaxis, number (%)**	31 (33.0)	37 (45.1)	NS
**UDCA prophylaxis, number (%)**	92 (97.9)	77 (93.9)	NS
**HDNAC prophylaxis, number (%)**	58 (61.7)	53 (64.6)	NS

BChE, butyrylcholinesterase (normal range for male 4620–11,500 U/L, for female 3930–10,800 U/L). Ferritin normal range 3.0–88.0 μg/L. IQR, interquartile range; UDCA, ursodeoxycholic acid; HDNAC, high dose N-acetylcysteine. * Fisher’s test. ** *U*-Test (Mann–Whitney).

**Table 5 jcm-08-00825-t005:** Association between minimum values of BChE activity and transplant outcomes.

Variables	BChE > 2000 U/L	BChE < 2000 U/L	*p*-Value *
**Patients, number (%)**	111 (63.1)	65 (36.9)	-
**Liver dysfunction, number (%)**	47 (42.3)	42 (64.6)	<0.05
**SOS, number (%)**	5 (4.5)	11 (16.9)	<0.05
**Allogeneic recipients, number (%)**	103 (58.5)	59 (33.5)	-
**Liver GVHD, number (%): ^#^**			
Absent	68 (66.0)	33 (55.9)	NS
I grade	12 (11.6)	4 (6.8)	NS
II grade	8 (7.8)	8 (13.6)	NS
III grade	10 (9.7)	7 (11.9)	NS
IV grade	5 (4.9)	7 (11.9)	NS
**Overall survival, number (%)**	94 (84.7)	38 (58.5)	<0.001
**Death, number (%):**	17 (15.3)	27 (41.5)	<0.001
Transplant-related	9 (8.1)	15 (23.1)	<0.05
Relapse	8 (7.2)	12 (18.5)	<0.05

BChE, butyrylcholinesterase (normal range for male 4620–11,500 U/L, for female 3930–10,800 U/L). SOS, sinusoidal obstruction syndrome. * Fisher’s test. ^#^ Percentages of liver GVHD calculated on the population of allogeneic recipients.

**Table 6 jcm-08-00825-t006:** Association between percentage drop of BChE values and transplant outcomes.

Variables	BChE Drop < 70%	BChE Drop > 70%	*p*-Value *
**Patients, number (%)**	94 (53.4)	82 (46.6)	-
**Liver dysfunction, number (%)**	53 (56.4)	36 (43.9)	NS
**SOS, number (%)**	2 (2.1)	14 (17.1)	<0.005
**Allogeneic recipients, number (%)**	86 (48.9)	76 (43.2)	-
**Liver GVHD, number (%): ^#^**			
Absent	57 (66.3)	44 (57.9)	NS
I grade	8 (9.3)	8 (10.5)	NS
II grade	8 (9.3)	8 (10.5)	NS
III grade	7 (8.1)	10 (13.2)	NS
IV grade	6 (7.0)	6 (7.9)	NS
**Overall survival, number (%)**	75 (79.8)	57 (75.0)	NS
**Death, number (%):**	19 (20.2)	25 (30.5)	NS
Transplant-related	9 (9.6)	15 (18.3)	NS
Relapse	10 (10.6)	10 (12.2)	NS

BChE, butyrylcholinesterase (normal range for male 4620–11,500 U/L, for female 3930–10,800 U/L); SOS, sinusoidal obstruction syndrome. * Fisher’s test. ^#^ Percentages of liver GVHD calculated on the population of allogeneic recipients.

**Table 7 jcm-08-00825-t007:** Association between patient transplant-related characteristics and transplant-related liver disease.

Variables	Severe Liver Damage	Liver Damage Absent or Mild	*p*-Value *
**Number of patients (%)**	44 (25)	132 (75)	-
**Underlying disease, number (%):**			
Acute lymphoblastic leukemia	18 (40.9)	47 (35.6)	NS
Acute myeloid leukemia	11 (25)	22 (16.7)	NS
Other	15 (34.1)	63 (47.7)	NS
**Disease risk, number (%):**			
Low	12 (27.3)	97 (73.5)	<0.0001
High	32 (72.7)	35 (26.5)	-
**Myeloablative conditioning, number (%)**			
MCHT-based	31 (70.5)	82 (62.1)	NS
TBI-based	13 (29.5)	50 (37.9)	NS
**Iron overload, number (%)**			
From absent to moderate	13 (29.5)	98 (74.2)	<0.0001
Severe	31 (70.5)	34 (25.8)	<0.0001
**Baseline BChE value, (U/L, median [IQR])**	6815 (4934–8640)	7655(5773–8827)	NS **
**Serum ferritin, (μg/L, median [IQR])**	2052 (1329–3112)	974 (329–2048)	<0.05 **
**Pre-transplant liver disease, number (%):**	9 (20.4)	28 (21.2)	NS
**Defibrotide prophylaxis, number (%)**	-	68 (51.5)	<0.0001
**HDNAC prophylaxis, number (%)**	25 (56.8)	86 (65.2)	NS
**Overall survival, number (%)**	19 (43.2)	113 (85.6)	<0.0001
**Death, number (%):**	25 (56.8)	19 (14.4)	<0.0001
Transplant-related	23 (52.3)	1 (0.8)	<0.0001
Relapse	2 (4.5)	18 (13.6)	<0.0001

Severe liver damage group included patients with SOS and liver dysfunction grade 3. BChE, butyrylcholinesterase; HDNAC, high-dose N-acetylcysteine. * Fisher’s test. ** *U*-Test (Mann–Whitney).

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
