# Peer review of "Drop of Butyrylcholinesterase Activity after Cyclophosphamide Conditioning as a Predictive Marker of Liver Transplant-Related Complications and Its Correlation with Transplant-Related Mortality in Pediatric Hematopoietic Stem Cell Recipients"

_jcm, 2019, doi:10.3390/jcm8060825_

Reviewer 1 Report

Maximova et al. invetigated the usefulness of serum BuChE  activity to predict liver complications after HSCT. They showed that although the baseline BuCHe activitiy  is poor a prognostic marker, reduction levels of BuChE activity by cyclophosphamide-treatment were strongly associated with post-tranplant liver damage. Overall, data this study contains some important insights. However, there are some issues as follows.

Title is sub-optimal. It shows only a part of this study,

There are no descriptions about the mesurements of biochemical markers in methods section. This point is very important.

Did the authors investigate whether the combinations of BuChE activity with other biochemical markers further improve the predictive performance?

L306-311, This part should move to Introduction section.

This manuscript should be proofread and edited by a native professionalEnglish expression is insufficient.

miner point

L29-30, 23.1% versus 8.1% do not mean lower values.

L53, This is not the first time to appear GVHD.

Author Response

We thank the reviewers for the opportunity to revise our manuscript titled “Drop of butyrylcholinesterase activity after cyclophosphamide conditioning for prediction of liver transplant-related complications in pediatric hematopoietic stem cell recipients.” We appreciate the careful review and constructive suggestions.  We have addressed the major concerns of the reviewers.

Review 1

1. Title is sub-optimal. It shows only part of this study.

We have changed the title according to the suggestion,emphasizing that the paper analyzes not only the correlation between BuChE reduction and the development of liver complications but also the transplant outcomes, with particular attention to Transplant Related Mortality.

2. There are no description about the measurements of biochemical markers in methods section.

As you suggest, we added a brief description of the measurements at the end of the methods section.

3. Did the author investigate whether the combination of BuChE activity with other biochemical markers further improve the predictive performance?

Preliminary data on the association between the drop in BuChE activity level and the markers listed in Table 2 did not show any statistical significance. For this reason, we preferred to exclude this data from the paper,giving prominence only to significant findings. According to the suggestion, we have added a brief explanation in the text.

4. L306-311. This part should move to introduction section.

We moved it.

5. This manuscript should be proofread and edited by a native professional English.

We sent our manuscript to a native English speaker for a new editing.

6. Minor points:

- L29-30, 23.1% versus 8.1% do not mean higher values

- L53. This is not the first time di appear GVHD.

Thank you, we made the needed correction.

We hope to hear from you.

Kind Regards,

The Authors

Reviewer 2 Report

This manuscript provides interesting and important findings, which may be useful for the prediction of transplant outcomes and the exploration of new treatments. However it seems that the diversity of the cohort is high, potentially reducing the impact of the study. I would like to recommend the authors to show the results when auto-transplant, nonmalignant disorders, and solid tumors are excluded as well.

Minor points.
Line #135:  The present study handles heterogenous data with a relatively long-period of the study. I would like to propose the authors to perform multivariate analysis, using variables such as disease risk, type of diseases, year of transplant, type of transplant, ABO mismatching and so on, to validate the impact of the cut-off value of BuChE on the transplant outcomes. ROS analysis using the selected cohort
that only includes patients with hematologic malignancies receiving allo-SCT. In that case, I would like to emphasize that it is not necessary to obtain p < 0.05.
Line #244: The cut-off value of BuChE was shown to be associated with TRM. Are there any data on causes of death according to the cut-off value?
Line #283: Fig 4 shows bimodal apparent increase of mortality between days 0-70 and days 150-250 in the BChE < 2000 group. What could the authors explain these phenomena?          

Author Response

We thank the reviewers for the opportunity to revise our manuscript titled “Drop of butyrylcholinesterase activity after cyclophosphamide conditioning for prediction of liver transplant-related complications in pediatric hematopoietic stem cell recipients.” We appreciate the careful review and constructive suggestions.  We have addressed the major concerns of the reviewers.

Review 2

1. I would like to recommend the authors to show the results when auto-transplant, non-malignant disorder and solid tumors are excluded as well.

Thank you. Following the suggestion, we added the results obtained when considering only patients with hematologic disease, analyzing both the development of liver damage and TRM. We are aware of the great diversity of our cohort. At the same time, the risk of post-transplant SOS appears to be substantially the same in the various group. For this reason we decided to analyse all the cases together.

2. …Multivariate analysis using variable such as disease risk, type of diseases, year of transplant, type of transplant and so on, to validate the impact of the cut-off value of BuChE on the transplant oucomes. ROS analysis using the selected cohort that only includes patients with hematologic malignancies receiving allo-SCT.

We added to the result section the results of our multivariate analysis performed using a Cox Proportional-Hazards Regression model.

3. Line 244. The cut-off value of BuChE was shown to be associated with TRM. Are there any data on causes of death according to the cut-off value?

We added the data regarding of cause of death in Results section.

4. Line 283: Fig 4 shows bimodal apparent increase of mortality between days 0-70 and days 150-250 in the BChE < 2000 group. What could the authors explain the phenomena?

This phenomena is due to the normal trend in transplant related mortality, with an early and a late form, as we have explained in the text.

We hope to hear from you.

Kind Regards,

The Authors

Round  2

Reviewer 1 Report

The authors have succeeded to revise the manuscript.